

# Offshore methane detection and quantification from space using sun glint measurements with the GHGSat constellation

Jean-Philippe W. MacLean [1], Marianne Girard [1], Dylan Jervis [1], David Marshall [1], Jason McKeever [1], Mathias Strupler [1], Antoine Ramier [1], Ewan Tarrant [1], and David Young [1]

[1]GHGSat Inc. Montreal, Qc, Canada, H2W 1Y5

**Correspondence:** jmaclean@ghgsat.com

**Abstract.** The ability to detect and quantify methane emissions from offshore platforms is of considerable interest to provide actionable feedback to industrial operators. While satellites offer a distinctive advantage for remote sensing of offshore platforms which may otherwise be difficult to reach, offshore measurements of methane from satellite instruments in the shortwave-infrared are challenging due to the low levels of diffuse sunlight reflected from water surfaces. Here, we use the GHGSat satellite constellation in a sun glint configuration to detect and quantify methane emissions from offshore targets around the world. We present a variety of examples of offshore methane plumes, including the largest single emission at $(84,000 \pm 24,000)$ kg h$^{-1}$ observed by GHGSat from the Nord Stream 2 pipeline leak in 2022 and the smallest offshore emission measured from space at $(180 \pm 130)$ kg h$^{-1}$ in the Gulf of Mexico. In addition, we provide an overview of the constellation's offshore measurement capabilities. We measure a median column precision of 2.1% of the background methane column density and estimate a detection limit, from analytical modelling and orbital simulations, that varies between 160 kg h$^{-1}$ and 600 kg h$^{-1}$ depending on the latitude and season.

## 1 Introduction

Methane is the second most important greenhouse gas after carbon dioxide (Pachauri and Meyer, 2014). Its short atmospheric lifetime, on the order of 10 years, has made it a key priority for reducing the rate of global warming in the near-term. While there has been a large focus on developing methane remote sensing technologies for onshore sites (Jacob et al., 2016, 2022), offshore targets remain an important and, until recently, understudied sector, which generates over a quarter of the total oil & gas production (IEA, 2018). With offshore natural gas production continually growing over the past two decades (IEA, 2018), it remains critical to develop cost-effective technologies for detecting and quantifying offshore methane emissions globally.

Offshore methane emissions have been measured using a variety of technologies including ship-based (Riddick et al., 2019; Yacovitch et al., 2020; Zang et al., 2020), aircraft (Gorchov Negron et al., 2020; Foulds et al., 2022; Ayasse et al., 2022; Gorchov Negron et al., 2023), and satellite measurements (Lorente et al., 2022; Irakulis-Loitxate et al., 2022; Roger et al., 2023). While drones and planes offer a solution for offshore targets near the coast, their limited range over water make continuous monitoring challenging. Untethered by these constraints, satellite remote sensing instruments offer a unique tool to frequently monitor methane emissions from offshore platforms anywhere in the world.



GHGSat operates a constellation of satellites for detecting and quantifying methane emissions. The technology demonstration satellite, GHGSat-D, launched in 2016, pioneered the use of high-resolution satellite images for detecting and quantifying methane emissions. Since then, GHGSat's commercial satellite constellation has grown to eight satellites that orbit the Earth in a sun-synchronous orbit at altitudes between 500 km and 535 km. The constellation uses a Wide Angle Fabry-Pérot (WAFP) spectrometer, operating in a narrow-band of the short-wave infrared (SWIR) spectrum to measure methane emissions over targeted domains of 150 $\mathrm{km}^2$ to 450 $\mathrm{km}^2$ with a pixel resolution of $\sim 25 \times 25$ $\mathrm{m}^2$. This high spatial and spectral resolution enables measurements of vertical column density of atmospheric methane with a column precision of 1.4-2.9% (interquartile range) of the background methane column density and a source rate detection limit of approximately 100 $\mathrm{kg\,h^{-1}}$ (McKeever and Jervis, 2021). In this way, GHGSat regularly detects and quantifies methane emissions from a variety of anthropogenic terrestrial sources: from oil & gas to hydroelectric reservoirs, coal mines, and landfills (Varon et al., 2019, 2020; Maasakkers et al., 2022; Cusworth et al., 2021).

In order detect and quantify methane emissions over land, the GHGSat constellation performs targeted measurements of sites with viewing angles within 20 degrees of nadir. However, for offshore measurements of methane, these types of targeted nadir-viewing observations are not optimal due to the low levels of SWIR radiation diffusely reflected from water surfaces. Nonetheless, the measured signal can be increased sufficiently to measure methane emissions when using a sun glint configuration. In this configuration, the satellite is oriented to align the target with the specular reflection of the sun off the ocean surface. A similar approach has been used by OCO2 to increase signal-to-noise levels at the detector by up to 3 orders of magnitude (Crisp et al., 2017) for measurements over oceans. Other missions have demonstrated the use of sun glint measurements for offshore methane detection and quantification. The TROPOMI instrument on the Sentinel-5p satellite mission used sun glint measurements to observe large scale variations in offshore methane column densities (Lorente et al., 2022). WorldView-3 and Landsat 8 measured offshore methane emissions from one platform in the Gulf of Mexico with an estimated source rate near 100,000 $\mathrm{kg\,h^{-1}}$ (Irakulis-Loitxate et al., 2022). The EnMap mission also used sun glint reflections to measure offshore methane emissions with source rates as low as 930 $\mathrm{kg\,h^{-1}}$ in the Gulf of Mexico (Roger et al., 2023). Having space-based remote sensing technologies that can approach the 100 $\mathrm{kg\,h^{-1}}$ detection limit offshore would allow for significantly more methane emissions to be monitored.

Here, we present GHGSat's capabilities for detecting and quantifying methane emissions from offshore platforms. We show a variety of detected methane plumes in offshore environments at different locations around the world and quantify their emissions rates. We evaluate the instrument performance by empirically measuring the column precision for an ensemble of glint observations taken in 2022 and spanning a range of viewing geometries. Finally, we build an analytical model based on empirical observations and orbital simulations to predict the detection limit of glint observations as a function of the target latitude and season.





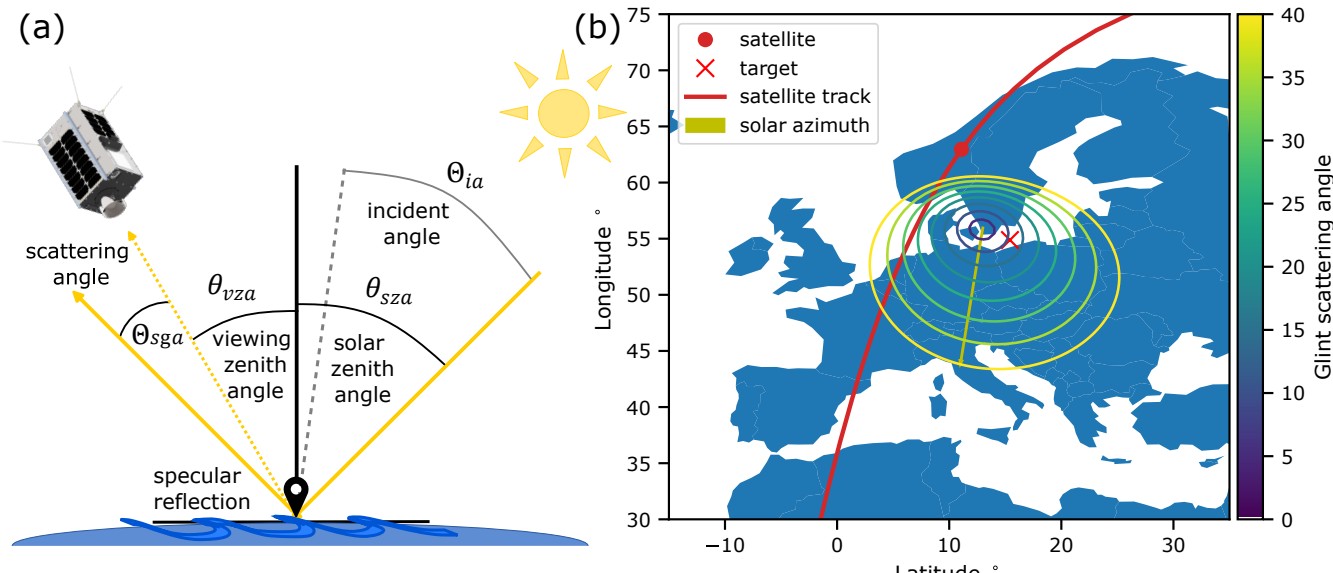

**Figure 1.** (a) Cross-section of the satellite viewing geometry for offshore sun glint observations. The satellite viewing zenith angle, $\theta_{vza}$, and solar zenith angle, $\theta_{sza}$, are measured with respect to the target normal (black solid line). Incoming solar radiation impinges on the water surface at incident angle, $\Theta_{ia}$, with respect to the wave surface normal (grey dashed line), and undergoes specular reflection towards the satellite. The scattering glint angle, $\Theta_{sga}$, indicates the angle separating the satellite view from the ideal direct solar specular reflection from a flat smooth water surface. (b) Example satellite ground track (red line) for viewing an offshore target (red cross) in the Baltic Sea. The solar zenith and solar azimuth angles (yellow line) for this example are 67.6 degrees and 189.4 degrees, respectively. Contour lines illustrate the glint scattering angle in intervals of 5 degrees from the satellite's perspective. The position of the satellite (red dot) indicates the position along-track that minimizes the scattering glint angle for viewing the target on this particular pass. Viewing this target requires a scattering angle of approximately 10 degrees.

## 2 Measurement overview

The measurement concept of operation is based on the wide angle Fabry-Pérot (WAFP) imaging concept as described in Jervis et al. (2021). The instrument measures back-scattered solar radiation in the SWIR around 1665 nm. The spectral radiance $L(\mathbf{x}, \lambda)$ reaching the instrument is calculated as

$$\mathbf{L}(\lambda, \mathbf{x}) = \frac{1}{\pi} R(\theta_{sza}, \phi_{saa}, \theta_{vza}, \phi_{vaa}, \lambda) \cos(\theta_{sza}) I(\lambda, \mathbf{x}), \tag{1}$$

where $R(\theta_{sza}, \phi_{saa}, \theta_{vza}, \phi_{vaa}, \lambda)$ is the surface reflectance function, $I(\lambda, \mathbf{x})$ is the spectral irradiance which includes the incoming solar irradiation and the atmospheric absorption with state parameter $\mathbf{x}$, and $\theta_{sza}, \theta_{vza}, \phi_{saa}, \phi_{vaa}$ are the solar and satellite viewing zenith and azimuth angles, respectively.

With targeted observations over land, and assuming a Lambertian surface, the surface reflectance reduces approximately to the spectrally dependent surface albedo, $R(\theta_{sza}, \phi_{saa}, \theta_{vza}, \phi_{vaa}, \lambda) \equiv a(\lambda)$. The spectral radiance at the instrument, away from spectral absorption features, therefore has a weak dependence on the satellite viewing direction. However, for offshore



measurements using glint mode, the reflectivity, $R(\theta_{sza}, \phi_{saa}, \theta_{vza}, \phi_{vaa}, \lambda)$, depends strongly on both the direction of incoming solar radiation as well the direction of the outgoing radiation towards the instrument.

For a given offshore target, we optimize the amount of light that reaches the detector by finding the satellite position along its track that minimizes the glint scattering angle (Capderou, 2014),

$$\Theta_{sga} = \arccos\left[\cos\left(\theta_{sza}\right)\cos\left(\theta_{vza}\right) - \sin\left(\theta_{sza}\right)\sin\left(\theta_{vza}\right)\cos\left(\phi_{saa} - \phi_{vaa}\right)\right], \tag{2}$$

with respect to the reflecting surface, as illustrated in Fig. 1(a). A scattering angle of zero corresponds to the satellite viewing the center of the glint spot: the direct specular reflection of the sun when the sea surface is flat. As such, the angle will be minimized when the solar and satellite zenith angles are equal, $\theta_{sza} = \theta_{vza}$ and the azimuths are offset by $\phi_{saa} - \phi_{vaa} = 180°$. The other angle that affects the signal reaching the detector is the incident angle,

$$\Theta_{ia} = \frac{1}{2}\arccos\left[\cos\left(\theta_{sza}\right)\cos\left(\theta_{vza}\right) + \sin\left(\theta_{sza}\right)\sin\left(\theta_{vza}\right)\cos\left(\phi_{saa} - \phi_{vaa}\right)\right], \tag{3}$$

which is the angle between the incident light ray and the sea surface normal, as illustrated in Fig. 1(a). Higher incident angles increase the surface reflectivity due to Fresnel reflection. Consequently, the signal levels at the detector can change significantly with the solar and satellite viewing geometry, in addition to other environmental conditions such as surface wind-speed and sea-surface roughness. We estimate the changes in surface reflectivity using the Cox-Munk sea-surface reflection model (Cox and Munk, 1954; Bréon, 1993; Bréon and Henriot, 2006),

$$R(\theta_{sza}, \phi_{saa}, \theta_{vza}, \phi_{vaa}, \lambda, ws) = \frac{\pi \rho_{fr}\left(\Theta_{ia}, \lambda\right)}{4\cos(\theta_{sza})\cos\left(\theta_{vza}\right)\cos^4(\beta)}P\left(Z_{up}, Z_{cross}, ws\right), \tag{4}$$

which depends on the slopes of the surface waves, $Z_{up}$, $Z_{cross}$, in the up-wind and cross-wind directions, respectively, the total wave surface slope, $\beta = \arctan\left(\sqrt{Z_{up}^2 + Z_{cross}^2}\right)$, and the Fresnel reflection coefficient, $\rho_{fr}$. $P\left(Z_{up}, Z_{cross}, ws\right)$ is the probability distribution function for a wave to have surface slopes $Z_{up}$ and $Z_{cross}$ given wind speed, $ws$, and for which Cox and Munk (1954) suggest using a Gram-Charlier decomposition. We calculate $P\left(Z_{up}, Z_{cross}, ws\right)$ from the satellite and solar angles $(\theta_{sza}, \phi_{saa}, \theta_{vza}, \phi_{vaa})$ using the method in Bréon and Henriot (2006).

While both the scattering angle and the incident angle change the total reflected signal, the scattering angle places the strongest constraints on which satellite passes can be selected to view a nearby target. Large along-track or cross-track angles are sometimes required to minimize the scattering angle in Eq. 2. Since the GHGSat satellites can view targets up to 65 degrees in the along-track direction and up to 55 degrees in the cross-track direction, this enables a larger number of opportunities which can achieve the scattering angle requirements for offshore measurements across a wide range of latitudes and throughout the year.

For each glint observation, the start time of the observation is obtained by propagating the satellite orbit and finding the time that minimizes the scattering angle in Eq. 2. We typically require the minimum scattering angle during an observation to be below 20 degrees as we have found empirically that scattering angles above this value considerably limit the light signal measured by the detector (see Fig. 4(a)). Figure 1(b) shows an example satellite ground track for a sun glint measurement of an offshore target with a scattering angle of approximately 10 degrees.



**Table 1.** Observation mode parameters for GHGSat constellation

| Parameter | Land | Offshore (glint) | Comments |
|---|---|---|---|
| Along-track angles | $\pm 20°$ | $\pm 65°$ | Used in nominal operations. |
| Cross-track angles | $\pm 20°$ | $\pm 55°$ | Used in nominal operations. |
| Maximum scattering angle | - | $20°$ | Used in nominal operations. |
| Satellite Target Distance | (500-535) km | (500-1550) km | Value depends on the satellite viewing zenith angle. |
| Ground sampling distance (GSD) | 25 m x 25 m | (25 m x 25 m)-(125 m x 125 m) | Value depends on the satellite viewing zenith angle. |
| Retrieval swath | 12 km | (12 - 60) km | Value depends on the satellite viewing zenith angle. |
| Revisit Opportunity Time | (7-14) days | (7-14) days | For one satellite. Depends on latitude. |

Once the observation begins, the satellites perform a panning manoeuver to acquire several closely spaced images as the target slowly pans through the field of view. Nominal observations consist of 200 frames with a frame period of 100 ms. The camera exposure time is adjusted for each observation based on the predicted reflectivity, obtained using the Cox-Munk sea-surface model, and the predicted wind-speed and wind-direction obtained from the meteorological database provided by the NASA Goddard Earth Observation System Fast Processing (GEOS-FP) reanalysis product. Table 1 summarizes the differences in the resulting observation parameters for land and offshore targeted observation modes.

In order to retrieve methane emissions in offshore environments, we use the differential absorption optical spectroscopy retrieval algorithm described in Jervis et al. (2021) and which has been validated for targets over land (Sherwin et al., 2023a, b). For each observation, we perform a (1) scene-wide retrieval using the full nonlinear forward model (FM) to determine scene-wide surface and atmospheric state parameters $\hat{x}$ followed by a (2) per-pixel column density retrieval using a linearized forward model (LFM) evaluated at the linearization point $\hat{x}$ to determine the surface and atmospheric state parameters at each ground cell.

The output of the column density retrieval produces a map of the state parameters for each ground cell in a reference coordinate frame as measured by the camera. The reference frame coordinate system is then georeferenced onto the appropriate UTM grid by leveraging computer vision reconstruction pipelines (Zhang et al., 2019) and using the satellite position and attitude at the reference frame trigger time. Camera distortion is corrected using pre-flight characterization data.

# 3 Example plumes

In Fig. 2, we show example retrievals where methane plumes were observed in a variety of offshore locations around the world. In each retrieval, we show the extracted plume overlaid on top of the surface reflectance using a floodfill masking algorithm. The source rates are estimated using the Integrated Mass Enhancement (IME) method as described in Varon et al. (2018). The $U_{10}$ wind speed is drawn from meteorological databases (GEOS-FP) after which the effective wind speed $U_{eff}$ is calculated using the method in Maasakkers et al. (2022). The error on the emission rate is obtained from the error on the wind speed, the error on the retrieved methane enhancements, and the error on the IME model, added in quadrature. We use a wind speed



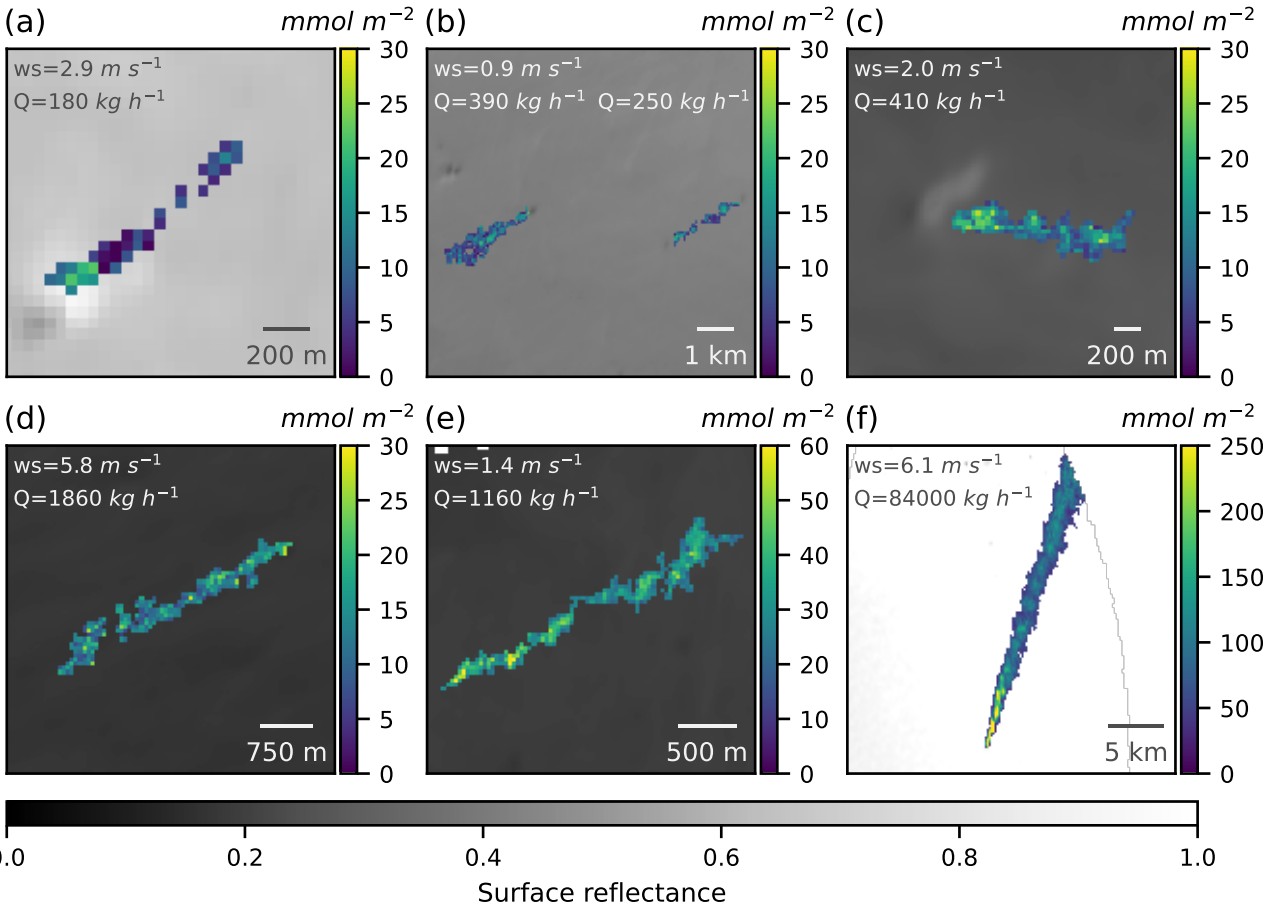

**Figure 2.** Retrieved offshore methane enhancement fields. The retrieved surface reflectance is plotted with the extracted methane plume overlaid on top. The plume masks show the retrieved methane column density enhancement above the local background in $\mathrm{mmol\,m^{-2}}$. (a-d) Examples from offshore shallow water platforms in the Gulf of Mexico for sites off the coast of Louisiana. Plumes measured from (a) a platform on Oct. 30th, 2022 with source rate $(180 \pm 140)\,\mathrm{kg\,h^{-1}}$, (b) two platforms on Feb. 5th with source rate $(390 \pm 300)\,\mathrm{kg\,h^{-1}}$ and $(250 \pm 190)\,\mathrm{kg\,h^{-1}}$, a central hub facility (c) on April 28th, 2022 with a source rate of $(410 \pm 230)\,\mathrm{kg\,h^{-1}}$ and (d) on Oct. 10, 2022 with a source rate of $(1860 \pm 820)\,\mathrm{kg\,h^{-1}}$. (e) Offshore platform off the coast of Africa measured on Nov. 24, 2022 with a source rate of $(1160 \pm 700)\,\mathrm{kg\,h^{-1}}$. (f) Nord Stream 2 pipeline leak in the Baltic Sea off the coast of Sweden, measured on Sep. 30, 2022 at 10:26 UTC with a source rate of $(84,000 \pm 24,000)\,\mathrm{kg\,h^{-1}}$.

error of $2\,\mathrm{ms^{-1}}$, obtained by comparing GEOS-FP with wind measurements at US airports (Varon et al., 2018), which is then propagated through the effective wind speed and source rate calculation.

Between October 2022 and April 2023, we observed a number of plumes at two sites in the Gulf of Mexico in shallow
waters off the coast of Louisiana [Figs 2(a-d)]. Shallow water platforms in the Gulf of Mexico have been found to exhibit





super-emitter behavior above $1000 \, \mathrm{kg \, h^{-1}}$ with high persistence (Ayasse et al., 2022) thereby increasing the carbon intensity of the region (Gorchov Negron et al., 2023). The source rates measured at these two sites range between $180 \, \mathrm{kg \, h^{-1}}$ and $1860 \, \mathrm{kg \, h^{-1}}$.

In Fig. 2(f), we show the observed methane emission from the Nord Stream 2 pipeline leak in the Baltic Sea. Following

reports of the Nord Stream 2 gas leaks on September 26th, 2022, GHGSat immediately began tasking its satellite constellation to detect and quantify methane emissions from the offshore Nord Stream 2 leak in the Baltic Sea. For the Nord Stream incident, while the first attempts by GHGSat and other satellites were unsuccessful due to cloudy conditions, three clear-sky opportunities on September 30th all yielded successful plume detections of the Nord Stream 2 pipeline leak. Figure 2(f) illustrates the largest of these plumes measured on Sep. 30, 2022 at 10:26 UTC at $(84,000 \pm 24,000) \, \mathrm{kg \, h^{-1}}$.

For the Nord Stream 2 plume, the wind speed error dominate the total error since the wind speed error of $2 \, \mathrm{ms^{-1}}$ is a substantial fraction of the predicted wind speed compared to the measurement or IME model errors. However, for the smaller plumes in Fig. 2(a-c), and the plumes in Fig. 2(d-e) with low albedo, the wind speed error is comparable to the error on the retrieved methane enhancements resulting in a total error that is a large fraction of the retrieved source rate. The quantification of the errors in the retrieved methane enhancements are discussed in detail in the next section.

These examples demonstrate the GHGSat constellation's capabilities for monitoring offshore platforms for a wide range of emission rates. With retrieved emission rates ranging from $84,000 \, \mathrm{kg \, h^{-1}}$ down to $180 \, \mathrm{kg \, h^{-1}}$, they represent, respectively, the single largest emission rate ever observed by GHGSat both on and offshore, and the offshore plume with the smallest emission rate detected from space to date.

## 4   Measurement performance

The error on the retrieved methane column density can be quantified by estimating the column density precision,

$$\Delta X_{CH_4}[mol \ m^{-2}] = \left(\mathbf{K}^T \mathbf{S}_o^{-1} \mathbf{K}\right)^{-1/2} \tag{5}$$

where $\mathbf{S}_o$ is the signal covariance error, which is calculated from the fit residuals during the column retrieve step, and $\mathbf{K}$ is the column retrieve Jacobian, which converts the signal error into an error on the retrieved parameters (Ramier et al., 2022).

We empirically measure the methane column density precision for an ensemble of 80 glint observations of offshore targets

at multiple locations around the world taken in 2022. For a uniform background, we expect retrievals of neighboring pixels to return approximately the same value. Deviations from this can therefore capture the noise in the measurement and the retrieval algorithm. Thus, the methane column density precision of a ground cell can be estimated by looking at the standard deviation of the retrieved methane column density in the surrounding ground cells.

For each observation, we compile a histogram of the standard deviation of a weighted moving filter with window size

$500 \, \mathrm{m} \times 500 \, \mathrm{m}$, weighted according to the number of valid ground cells in the window. Ground cells are rejected if they have a surface reflectance below 0.04 or a posterior methane error, calculated using Eq. 5, above $0.030 \, \mathrm{mol \, m^{-2}}$. The total histogram for the 80 observations is shown in Fig. 3(a). We find a median column density precision of 2.1% ($13.5 \, \mathrm{mmol \, m^{-2}}$)



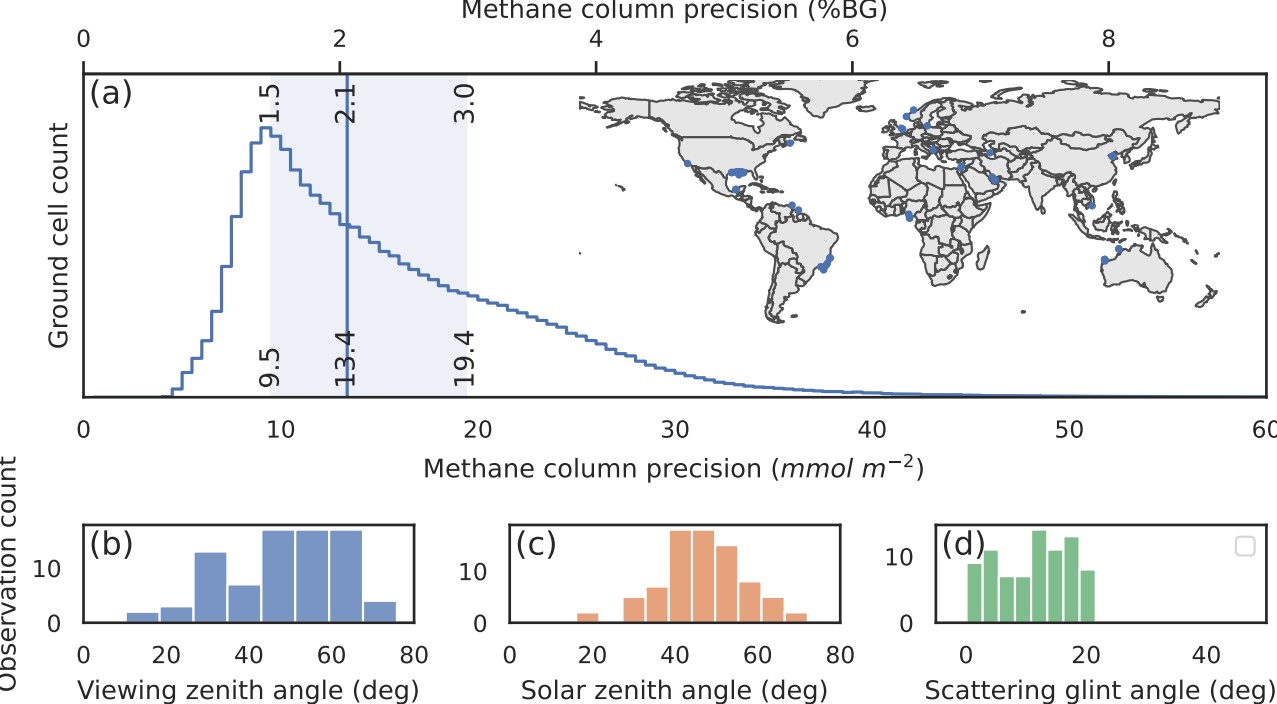

**Figure 3.** Methane column density precision for offshore glint measurements. (a) Histogram of the standard deviation of the retrieved methane column densities for a moving window with size 500 m × 500 m for 80 observations taken in 2022. We find that ground cells between the 25th and 75th percentiles have a standard deviation between 1.5 and 3.0% of the methane background column density, with a median value of 2.1% of background. Inset: Target locations for the observations used in the analysis. Histograms of the (b) viewing zenith angle, (c) solar zenith angle, (d) scattering zenith angle for the observations analyzed. Measurements were performed over a wide range of solar and satellite viewing conditions, with the majority of observations having solar and viewing zenith angles between 20 and 70 degrees. Only observations with a glint scattering angle below 20 degrees are used in the analysis (see text for details).

of the background methane column density, with an interquartile range between 1.5% and 3.0% (9.6 and 19.6 $\mathrm{mmol\,m^{-2}}$). The median value at 2.1% is comparable to the observed value on land at 2.0% (Ramier et al., 2022). On the lower end, we find

a cutoff where all 500mx500m ground cells have a column density error above 0.8% of background. The long tail extending beyond 3.0% is primarily due to observations with low signal levels, as discussed below.

We can estimate how the column density precision scales in the shot-noise limit. In this case, the signal covariance is dominated by the photon shot-noise, $\mathbf{S}_o \propto \sigma_N^2 \propto I$, which is proportional to the measured signal, $I$. Moreover, the normalized methane Jacobian, $\mathbf{k}_{CH_4} = \mathbf{K}_{CH_4}/I$, can be shown to be approximately proportional to the airmass factor, $\mu(\theta_{sza}, \theta_{vza}) =$

$1/\cos\theta_{sza} + 1/\cos\theta_{vza}$, which accounts for the total path traveled by a light ray. If we assume the main contributor to the



Jacobian are the column elements involving methane, $\mathbf{K} \approx \mathbf{K}_{CH_4}$, then, we find,

$$\Delta X_{CH_4}[mol\ m^{-2}] \approx \frac{\sigma_N}{I\left(\mathbf{k}_{CH_4}^T \mathbf{k}_{CH_4}\right)^{1/2}} \approx \frac{\alpha}{\mu\sqrt{I}} \propto \frac{\alpha}{\mu\sqrt{R\left(\theta_{sza},\theta_{vza},\phi_{saa},\phi_{vaa},\lambda\right)\cos(\theta_{sza})}}, \tag{6}$$

where $\alpha$ is a proportionality constant to be determined empirically.

For the case of solar illumination, the measured signal, $I$, is proportional to the surface reflectance, $R$, and the cosine of the solar zenith angle $\theta_{sza}$, $I \propto R\cos(\theta_{sza})$. For unpolarized light, the reflectance can be parametrized by two angles, the scattering angle, $\Theta_{sga}$, and the incident angle, $\Theta_{ia}$. These two angles will vary with each glint observation depending on the relative positions of the sun, target, and satellite, changing the signal reaching the instrument. The measured mean signal for an ensemble of glint observations is shown as a function of these two angles in Fig. 4(a-b). In Fig. 4(a), we find the mean signal decreases linearly with the scattering angle as the satellite gets further and further away from the optimal glint spot. Beyond a glint scattering angle of 20 degrees, the mean signal falls below $30\ \mathrm{ke\,s^{-1}}$ where the signal can be too low to retrieve methane. In Fig. 4(b), we see that higher incident angles increase the mean signal due to increased Fresnel reflection.

As the signal levels vary with viewing geometry, the column density noise will change. In Fig. 4(c-d), we further break down the measured column precision from Fig. 3 by comparing it against different retrieved parameters. In Fig. 4(c-d), we plot the mean column precision of each observation as a function of the retrieved mean surface reflectivity [Fig. 4(c)], and the scaling factor $1/\mu\sqrt{I}$ from Eq. 6 [Fig. 4(d)]. In Fig. 4(c), we find the mean column precision is inversely proportional to the surface reflectivity; a higher surface reflectivity leads to a higher light signal at the detector, and this translates into a reduction in random errors. Above an albedo of 0.3, the retrieved median methane error remains below 2% of the background whereas below this value, it increases steadily to 4-5% of background at an albedo of 0.1. In Fig. 4(d), we find the mean column precision to be linearly proportional to the scaling factor, $1/\mu\sqrt{I}$, with slope $\alpha = 0.288\ \mathrm{mol\,m^{-2}(kes^{-1})^{-1}}$ and intercept $0.003\ \mathrm{mol\,m^{-2}}$. The variations in the methane column precision can therefore be estimated by the change in airmass factor and signal levels in Eq. 6 over the large range of solar and viewing angles under study.

## 5  Detection limit for offshore measurements

The detection limit defines an instrument's ability to detect plumes. For observations on land, the detection limit of GHGSat's satellite remote sensing instruments has been assessed with controlled release experiments, either through internally-organized campaigns within GHGSat or single-blind campaigns organized with third parties (Sherwin et al., 2023b; Darynova et al., 2023). However, such experiments are extremely challenging in offshore environments for both operational and regulatory reasons, and, as such, we assess the glint instrument detection limit through a combination of the empirical measurement results presented in the previous section and analytical modelling.

In the limit of a single pixel detection, the methane detection limit of an instrument is driven by the following equation (Jacob et al., 2016),

$$Q_{lim} = M_{CH_4}\ U\ G\ q\Delta X_{CH_4} \tag{7}$$





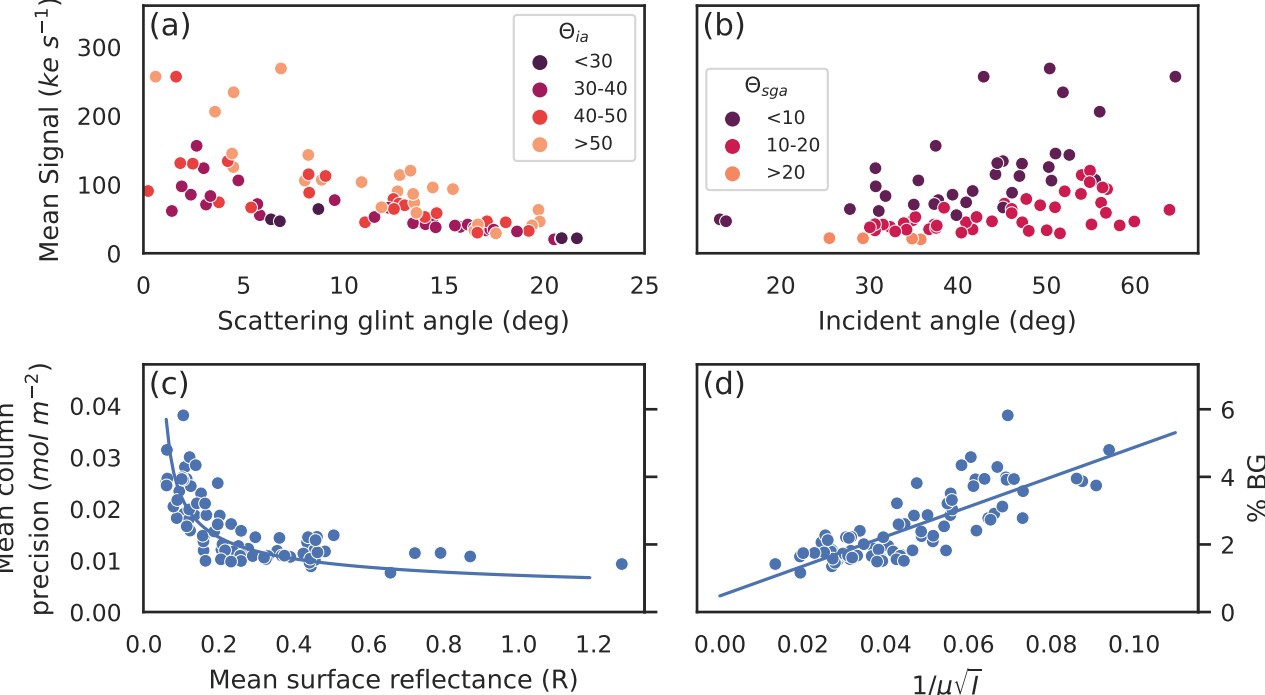

**Figure 4.** Relationship between mean viewing geometry, signal, and methane column precision for 80 glint observations taken in 2022. (a-b) Effect of the viewing geometry on the mean measured signal. The observation mean signal ($\text{kes}^{-1}$) decreases with the (a) scattering angle and increases with (b) the incident angle. As the scattering angle increases, the satellite view moves away from the ideal glint spot and the signal decreases, whereas as the incident angle increases, the reflected light at the surface increases due to Fresnel reflection. (c-d) Effect of the change in the signal on the mean column precision for the same observations. (c) The observation mean column precision is found to be inversely proportional to the mean surface reflectance. Above a surface reflectance of 0.3, we find the methane column density to be fairly constant near 2% of the methane background column density. Below this value, it increases to 4-5% of the background. (d) The mean column precision is found to be linearly proportional to $1/\mu\sqrt{I}$ from Eq. 6, the inverse of airmass factor times the square-root of the mean signal.

where $M_{CH_4}$ is the molar mass of methane, $U$ is the wind speed, $G$ is the ground sampling distance (GSD), $\Delta X_{CH_4}$ is the methane column precision, and $q$ determines the number of standard deviations above the noise required to reliably assign a methane enhancement to a pixel. Values of $q = 2$ and $q = 5$ have been proposed for detection and quantification, respectively (Jacob et al., 2016). GHGSat's instruments operating in targeted mode over land have a ground sample distance of 25 meters and a median methane column density noise of 0.013 $\text{mol m}^{-2}$, or 2% of the background column density. Assuming a wind speed of 3 m/s and a value of $q = 2$ for detection, Eq. 7 gives a detection limit of 120 $\text{kg h}^{-1}$, a value commensurate with results obtained from controlled release experiments on the ground (McKeever and Jervis, 2021; Ramier et al., 2022).

For land observations operated in nadir mode over diffuse surfaces, the GSD and column precision vary little with viewing angle, and, as a result, the detection limit remains fairly constant across different observations. With glint observations, how-



ever, both the ground sample distance and column density noise can vary significantly for different viewing geometries and these affect the detection limit of an observation. In order to estimate the detection limit with glint observations, we approximate the GSD,

$$G = \frac{h_{sat} p_{pix}}{f} \frac{1}{\cos(\theta_{vza})^{1/2}}, \tag{8}$$

as the ratio of the satellite target distance, $h_{sat}$, the pixel pitch, $p_{pix}$, and the instrument focal length $f$. The satellite target distances can be expressed in terms of the satellite's nadir altitude $h_0$, its viewing zenith angle, and the Earth's radius $R_E$, $h_{sat} = \sqrt{R_E^2 \cos^2(\theta_{vza}) + h_0(h_0 + 2R_E)} - R_E \cos \theta_{vza}$. The inverse cosine with exponent 1/2 is obtained from the geometric mean of the GSD along the two camera axes, one which increases only with the satellite-target distance, $h_{sat}$, and the other which increases due to both the satellite target distance and pixel projection effects, $h_{sat}/\cos(\theta_{vza})$. Furthermore, Eq. 7 and

Eq. 8 can provide an initial estimate of the expected range for the glint detection limit. Fixing the methane column density precision, $\Delta X_{CH_4}$, at the measured median value in Fig. 3(a), viewing zenith angles between 20 and 70 degrees from Fig. 3(b), and assuming winds speeds of $3 \text{ ms}^{-1}$, we find a detection limit of approximately $135 \text{ kg h}^{-1}$ at 20 degrees and $500 \text{ kg h}^{-1}$ at 70 degrees.

To take into account the variations in the methane column density precision, we substitute Eq. 6 and Eq. 8 into Eq. 7. We
find a detection limit that scales as,

$$Q_{lim} \approx \alpha M_{CH_4} U \frac{h_{sat} p_{pix}}{f \cos(\theta_{vza})^{1/2}} \frac{q}{\mu \sqrt{I}}. \tag{9}$$

Equation 9 connects the detection limit of an observation to the measured signal level and viewing geometry. By simulating how the signal varies for different viewing geometries, target latitudes, and times of year, we can estimate the glint detection limit of the GHGSat constellation.

## 6    Detection limit from orbital simulations

To estimate the detection limit, we propagate a series of sun-synchronous satellite orbits at an altitude of 535 km and with a local time at the descending node (LTDN) between 10h00 and 14h00 in 1 hour steps for a period of 60 days around the summer/winter solstices and the spring/fall equinoxes: December, March, June, September 2022. We simulate observations of glint targets at latitudes between -60 and 60 degrees and at these four different times of year. For each target observation,
we select the observation start time by minimizing the scattering angle in Eq. 2. Satellite passes are deemed valid when the minimum scattering angle is below 20 degrees and the satellite viewing zenith angle is below 80 degrees. In this way, the total number of observations of each target varies between 30 and 180 depending on the latitude.

For each valid observation, we tabulate the solar and satellite zenith and azimuth angles at the observation center. The distribution of viewing zenith angles for each latitude and season is illustrated in Fig. 5. During the equinoxes (March/September),
the viewing zenith angle across all simulated observations is small near the equator whereas during the solstice (June/December) the viewing zenith angles is small near the Tropics at $\pm 23.5$ degrees. At higher latitudes, the viewing zenith angle increases



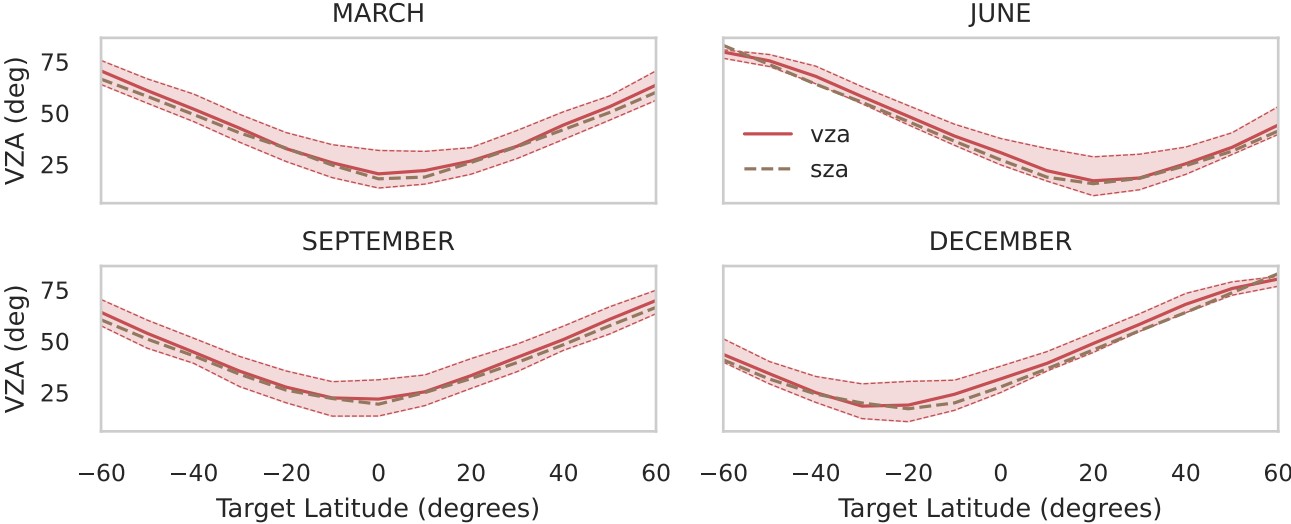

**Figure 5.** Satellite viewing zenith angle and solar zenith angle for the ensemble of simulated satellite observations at different latitudes and seasons. Dashed and solid red lines show the 25th,50th, and 75th, viewing zenith angle percentiles for the ensemble of observations at each latitude. The dashed black line illustrates the median solar zenith angle for each latitude. For each satellite pass, the observation center is obtained by minimizing the scattering angle, $\Theta_{sga}$. We find that the optimal viewing zenith angle for the ensemble of observations tracks the solar zenith angle as a function of latitude and season. At high latitudes, when the sun is closer to the horizon (large SZA), the satellite is constrained to stay close to the horizon (large VZA) to minimize the glint scattering angle.

when the solar zenith angle is correspondingly large. We thus find that for glint observations to minimize the scattering angle, the satellite viewing zenith angle will, on average, track the solar zenith angle throughout the year.

The solar and satellite angles are then used to calculate the predicted signal, $I$, and, second, to determine the methane column

precision, $\Delta X_{CH_4}$. The predicted signal is calculated from an empirically measured instrument transmission function and the surface reflectance using the Cox-Munk model (Bréon and Henriot, 2006), assuming a fixed wind speed of 3 m/s and averaged over four wind directions, 0, 90, 180, and 270 degrees from North. Once the signal is calculated, we estimate the column precision from the signal-noise relationship following Eq. 6 and the empirically measured proportionality constant obtained in Fig. 4(d).

The detection limit is then calculated using Eq. 7, using a value of $q = 2$, and the results are presented in Fig. 6. Solid and dashed lines show the detection threshold percentiles for the ensemble of observations at each target. We find the detection threshold varies with latitude and season. During the summer solstice, the range of detection thresholds is lowest in the Tropics near $\pm 23.5$ degrees whereas during the equinoxes in March and September, the range of detection thresholds is lowest near the equator. We find an increase in the detection limit at higher latitudes for all seasons. The increase is larger in winter solstice,

which corresponds to northern latitudes in December and to southern latitudes in June. We summarize these findings in Table 2



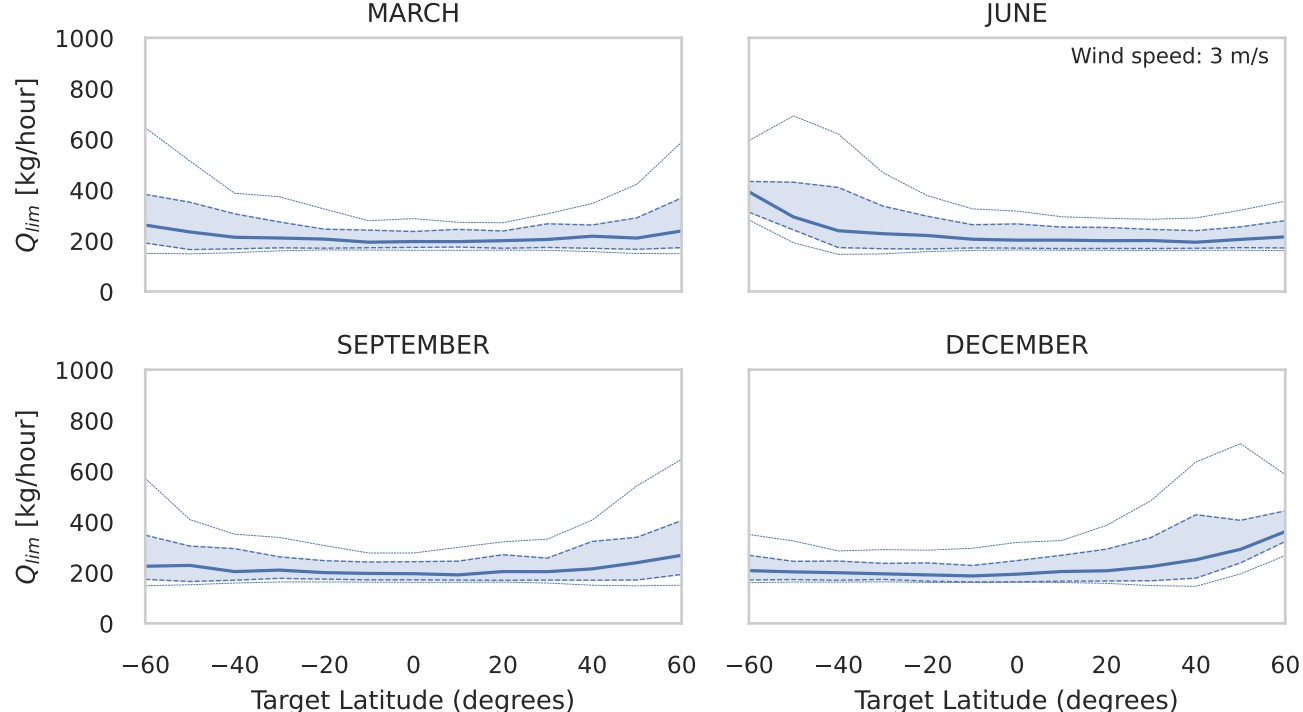

**Figure 6.** Estimated detection threshold as a function of latitude for spring/fall equinoxes and summer/winter solstices. For each period and target latitude, we simulate 60 days of satellite observations for sun-synchronous orbits at 535 km with LTDN between 10h and 14h. Observations with a scattering glint angle below 20 degrees are retained. The column density noise and GSD are calculated from the predicted signal based on the solar and satellite viewing angles. Dashed and solid lines show the 5th,25th,50th, 75th, and 95th detection threshold percentiles for the ensemble of observations at each latitude. Filled values show the range between the 25th and 75th percentiles. We find that the detection threshold is lower at latitudes and times of the year when the sun is higher in the sky - near equatorial latitudes in the summer and near the Tropics in the spring and fall.

which shows the mean detection limit range for two latitudes bands and the four seasons. The values in the table for the 5th and 95th percentiles vary between $160\,\mathrm{kg\,h^{-1}}$ and $600\,\mathrm{kg\,h^{-1}}$.

The variations in the detection limit in Fig. 6 can be understood by comparing it to the variations in the viewing zenith angle in Fig. 5. The viewing zenith angle affects both the GSD and the methane column precision which in turn affects the detection limit. Observations with larger viewing zenith angles will have a larger satellite target distance, and consequently a larger GSD, which will increase the detection limit. Larger viewing angles will however decrease the methane column density noise due to an increase in both the signal level reaching the instrument through larger Fresnel reflection, and an increase in the airmass factor, $\mu$. While the decrease in the methane column density noise will decrease the detection limit, this effect is small compared to the effect of the GSD. As such, the changes in the detection limit in Fig. 6 are approximately correlated with the changes in the viewing zenith angle in Fig. 5. These results illustrate the main difference between glint observations and



**Table 2.** Detection limit range estimated from simulated glint (offshore) measurements for two latitude bands. Values represent quantiles in $\mathrm{kg\,h^{-1}}$ averaged over each specified latitude band and season in Fig. 6 rounded to the nearest ten.

| | Summer | | | Spring/Fall | | | Winter | | |
|---|---|---|---|---|---|---|---|---|---|
| | | | | | Quantile | | | | |
| Latitude (degrees) | 0.05 | 0.25-0.75 | 0.95 | 0.05 | 0.25-0.75 | 0.95 | 0.05 | 0.25-0.75 | 0.95 |
| 0-30 | 160 | 170-250 | 300 | 160 | 170-250 | 300 | 160 | 170-290 | 380 |
| 30-60 | 160 | 170-250 | 310 | 160 | 170-280 | 380 | 170 | 200-400 | 600 |

targeted land observations, namely, that the viewing conditions for each glint observation play an important role in determining the detection limit, and these vary with latitude and season.

## 7 Conclusions

We have demonstrated the capability for satellite-based remote sensing to detect and monitor both large and small offshore
methane leaks in near real-time with the GHGSat constellation. Detection and quantification of offshore methane emissions is operational now and has been demonstrated with multiple examples including the smallest offshore emissions measured from space to date.

Glint observations are unique in that the viewing conditions can change with every observation and this affects the signal levels, the methane column density precision, and the ground sample distance (GSD). We empirically measured a median column
precision of 2.1% of the background methane column density, a value comparable to what we obtain on land. Furthermore, by combining an analytical model of the detection threshold with empirical measurements of the column precision, we find a detection limit that can vary between (160-600) $\mathrm{kg\,h^{-1}}$ depending on the target latitude and time of year of the observation. The detection limit is predicted to be better at lower latitudes and in summer months when the solar zenith angle is small. We note that while the detection limit values are specific to the GHGSat constellation, the formulas and scaling relationships
obtained are general and would apply to any satellite measurement of atmospheric gases in a glint configuration.

The analysis presented applies to an ensemble of simulated orbits for a few specific targets. In practice, with point source imagers, a satellite operator must choose to observe select targets from a large ensemble on any given day. It should therefore be possible to optimize the selection of targets and the selection of the satellite pass to view those targets based on criteria that include minimizing the detection limit. Moreover, the analytical calculation of the detection limit assumes single pixel
detection. However, with point source imagers, multiple pixels are typically required to confirm a detection. While this may change the detection limit for a given latitude and season, we don't expect this to affect the scaling relationships. Further work would be required to understand the impact of requiring multi-pixel detection when estimating a detection limit. Ultimately, controlled release experiments are necessary to validate the glint detection limit. However, due to operational and regulatory complexities of offshore controlled release experiments, simultaneous satellite measurements with aircraft or other ground
based instruments across many locations may be a good alternative.



With three additional satellites planned for launch by the end of 2023, the GHGSat satellite constellation will have 10 operational satellites in orbit for detecting and quantifying methane emissions. With an increasing number of satellites and as more data is collected, the detection limit model can be refined, and, eventually, validated through controlled releases or cross-validated with other instruments. The methods and analysis presented here will also be applicable to the upcoming $CO_2$
demonstration satellite to be launched at the end of 2023. As the constellation continues to grow, this will enable the detection, quantification, and, ultimately, the mitigation of methane emissions from any site, on and offshore, with near daily revisit opportunity times.

*Author contributions.* JPWM and JM developed and implemented the glint observation mode concept and operationalized the measurement. JPWM, JM, and DJ developed the analytical model. JPWM and AR wrote the data analysis software. JPWM performed the orbital
simulations, analyzed the empirical and numerical data, and wrote the manuscript with comments and revisions from all authors.

*Competing interests.* The authors declare that they have no conflict of interest.

*Acknowledgements.* We acknowledge the support of the Canadian Space Agency (CSA) [22AO2 - C - CO]. Furthermore, we would like to thank Space Flight Laboratory and C-Core for the close collaborations in operationalizing glint observations.



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
