# Peer review of "Offshore methane detection and quantification from space using sun glint measurements with the GHGSat constellation"

_EGUsphere, 2023_

## Author Response (AR1)

**Referee #2**

*This work focuses on the detection and quantification of methane emissions at offshore areas using the GHGSat constellation data. The advanced GHGSat technology allows the detection of very weak plumes with flux rate values of hundreds of kg/h lower than the lowest emission detected from space to date. Because of the offshore oil and gas industry accounts for approximately a third of the total oil and gas industry, it is important to monitor methane emissions from offshore areas.*

*In this study, a detailed detection limit analysis based on a comprehensive sampling and simulations has been carried out. It relates the detection limit to the target latitude, time of the year, and different angular configuration parameters. This analysis concluded that the GHGSat constellation presents a general detection limit lower than 1 t/h for offshore areas, which leads to an unprecedented detection capability for space-based instruments.*

*The manuscript presents remarkable writing, an adequate presentation of results, and it is the result of exhaustive research. I congratulate the authors on this.*

*I found three main points for clarification and some minor corrections.*

*Major corrections (points for clarification)*

*- Did you check if the predicted GEOS-FP wind speed value matched the recalculated GEOS-FP value? Can you trust the predicted value to predict the reflectivity? Please, provide some clarification in the manuscript.*

> It is unclear what the referee is referring to with "the recalculated GEOS-FP value".
>
> We use the GEOS-FP wind speed forecast generated before the observation time only for setting the camera gains on the instrument at the time of the observation.
>
> Once the observation has been taken and downlinked, we use the GEOS-FP wind speed reanalysis product calculated at the observation time. This historical wind speed is used for source rate quantification as described in the text.
>
> This distinction has been clarified.

*- Besides, in L135-136 I understand that the predicted GEOS-FP wind speed value is the wind speed value used for quantification. Why not using the recalculated GEOS-FP value? Are predicted and recalculated values so close that there is practically no difference? Please, provide some clarification in the manuscript.*

As in the previous question, it is unclear what the referee is referring to with "the recalculated GEOS-FP value".

The GEOS-FP wind speed reanalysis product is used as an input for source rate quantification. This approach has been validated in multiple controlled release experiments, see *Sherwin et al.* 2023.

This distinction has been clarified in the text.

*- L162-163. Precision is retrieved assuming that all the pixels are above the shot-noise limit. Is this true? If not, can we trust the 2.1% median precision? Please, provide some clarification in the manuscript.*

The column precision is estimated by taking the standard deviation in the retrieved methane enhancement for valid non flagged ground cells in a 500 x 500 m window.

We do not require any assumption about the ground cells being above the shot noise limit. The 2.1% median column precision applies to all non-flagged pixels which are the relevant ones used for source rate quantification.

We have provided more clarification on the quality flags that we apply.

*Minor corrections*

*- L36: In order TO detect…*

Updated

*- L46-47: 'Gulf of Mexico' is repeated. The second sentence can be reformulated to avoid repetition.*

*The second "Gulf of Mexico" has been removed.*

*- Figure 1: Figure 1b could be explained more plainly in the caption.*

The caption has been reformulated.

*- L117: I would suggest more information about the masking process because its importance for quantification.*

Additional details on the floodfill algorithm have been added.

*- Figure 2: I suggest to show retrievals with no masking of at least a couple of cases (weak and strong plumes) for the reader to trust the masking.*

We have added a figure to the appendix with the retrieved surface reflectance and unmasked methane enhancement fields for sample observations found in Figure 2.

*- L156-157: why rejecting surface reflectance values < 0.04 or methane errors > 0.03? Please, provide some clarification in the manuscript.*

As standard practice, GHGSat produces a quality flag layer with its retrieval to indicate ground cells where the retrieved state parameters may be less reliable. Ground cells are flagged when the reflectance is either too low (below 0.04) or the calculated posterior error is too high (above 0.03 mol/m^2). These values are chosen to balance the number of pixels retrieved (we want this to be large) against the measurement error (we want the scene to have minimal artefacts). Ground cells which are flagged are not used in the subsequent source rate quantification.

Clarification has been added in this regard.

*- L160: '500 m x 500 m' (separation)*

Updated.

*- 'Measured signal' and 'Irradiance' have the same symbol (I). I would suggest to change one of them, e.g., Irradiance = Irr.*

We have changed the symbol for spectral irradiance in the manuscript.

*- L162-168: I think that there is an abrupt evolution from Eq.5 to Eq.6. I would suggest a more extended explanation to better understand how the precision formula is deduced.*

Eq. 6 and the preceding paragraph have been modified to make the connection to Eq. 5 more clear.

*- L265: offshore emissions can also come from inefficient flaring, for example, not only leaks.*

The wording has been changed.

We thank the referee for their comments and suggestions which we believe have strengthened the manuscript.

**Referee #3**

*The work presented by MacLean et al. is relevant, proving that offshore measurements from the GHGSat constellation of satellites is possible. This comes in a timely moment as methane is in the spot for regulations and stakeholders are being more aware of the methane monitoring technology. The manuscript discusses various aspects of the measurements, retrieval algorithm and detection and quantification of methane plumes offshore. The text is well written and succinct, with clear figures (and no typos which is appreciated from a reviewer's perspective). However, the manuscript lacks some details regarding the methodology used, as it directly references the original sources without providing sufficient detail. While it's possible that the assumption of reader familiarity with various methodologies has been made by the authors, it shouldn't be taken for granted. The manuscript would be more comprehensive with the inclusion of additional details. Furthermore, more in-depth discussion on the choices and assumptions made and their implications to the retrievals and emission quantification are missing. Including those will make those assumptions stronger and the manuscript more scientifically sound. Therefore, I recommend publication after proper revisions have been made. See below for more specific comments:*

**Main points:**

*Line 105-110 about the retrieval algorithm: readers will appreciate some more details on the retrieval algorithm without needing to go to the main reference. Is there any relevant difference between the land and ocean retrieval algorithm, e.g., state vector? Is this a physics-based retrieval? How sensitive is to scattering processes in the atmosphere, are those taken into account?*

> While the way offshore observations are made differs from the land observations, the retrieval algorithm is the same. We use a physics-based retrieval algorithm as described in Jervis et al. 2021 with an identical state vector. Additional details have been provided in this regard.

> Scattering is neglected in the retrieval algorithm. As described in Jervis et al, 2021, GHGSat retrievals are primarily intended to measure local plume enhancements and errors arising from neglecting scattering should be small compared with other sources of measurement error.

*Limit of 20 deg for the scattering glint angle. From Fig. 4 angles between 15-20 doesn't seem to provide high levels of signals neither, also for incident angles lower than 30. In Line 175: 'signal can be too low' reads very vague. More discussion on the choice of 20 degrees and how this affects precision should be added.*

> Beyond 20 degrees, we have found that there is often not enough light reaching the detector to perform a reliable measurement. The signal to noise level is too low, and as

such, we are unable to retrieve methane enhancements in the scene nor attribute a column precision. We have modified the sentence in the paragraph to make this clear.

Both the incident angle and scattering angle matter, but the signal is more sensitive to a change in the scattering angle. It is however possible to have an incident angle below 30 degrees with enough signal as the two data points below this value in Fig. 4b illustrate.

*There is not a single image of the complete targeted domain. How noisy it is, does it suffer from artifacts related to atmospheric conditions, for example aerosols, wind speed conditions? What about false positives? (look in the control release manuscripts what is that GHGSat suffers more from).*

We have added a figure to the appendix with the retrieved surface reflectance and unmasked methane enhancement fields for sample observations and over a larger domain than what is shown in Fig 2.

*Figure 2 and paragraph starting in line 135: it would be interesting to include some discussion on the difference in the surface reflectance. Is the reflectance shown in the figure the modelled surface reflectance in the retrieval, or the retrieved one? Or is this estimated by the floodfill masking algorithm (line 117, which would be good to shortly define as well).*

The paragraph starting on line 135 has been rewritten to include a discussion of the relative contributions of the surface reflectivity and wind speed to the total error.

The surface reflectance shown is the retrieved surface reflectance as described in the caption of Figure 2. This clarification has been added to the text.

A short description of the floodfill algorithm has been added.

*Figure 2 as well. It is mentioned that in the Nord stream case the assumed wind speed error of 2m/s is a substantial fraction of the predicted wind speed. Is this not the case for the other plumes where wind speed is even smaller?*

For the Nord Stream 2 plume, the retrieved surface reflectivity (albedo) is much higher leading to a smaller relative error in the retrieved column densities compared to the smaller plumes. The wind error is still important in all case, but the relative contribution of the wind error is larger for the Nord Stream 2 plume. Clarifications have been added.

*Precision estimate:*

*-Do all the 80 observations contain XCH4 enhancements, or are also methane free scenes included? How sensitive is the precision estimate to the choice of 500m x 500 m window? It would be good to see how this window looks like in a complete scene, as in the manuscript only the masked plumes are shown.*

These are 80 observations where no methane plumes were detected. This has been clarified in the text.

We chose a 500 x 500 m window as this is approximately the scale over which plumes are detected. As the window size increases, the standard deviation in the retrieved methane enhancement does increase slightly since more ground cells are included. However, as we are interested in the column precision on length scales which affect plume detection, a 500 m x 500 m window seemed appropriate.

Moreover, we have added the unmasked plumes to the appendix.

*-In addition to the 0.04 surface reflectance and methane error of 0.03 mol/m2, are there any other rejection criteria applied, e.g., based on measurement geometry?*

Geometric considerations are only taken into account when deciding whether or not to take an observation by calculating the glint scattering angle. They are not considered for filtering individual ground cells.

*-Figure 4 and discussion in the text: given that the median error is 0.135, it would be good to draw a horizontal line at this level. I understood from the text that observations with precision higher than 0.03 are rejected, but I see in this Figure points higher than that. Furthermore, looking at Fig. 4c I wonder if the median value 2.1% is the most representative one.*

We have modified Fig. 4 to include the 25$^{th}$, 50$^{th}$, and 75$^{th}$ percentiles from Fig. 2.

The posterior methane error, calculated with Equation 5, is what the forward model predicts the error to be. We use this to reject pixels which are not included in the source rate quantification. This is different than the empirically estimated column precision illustrated in Fig. 4. The empirically estimated metric can include sources of errors which are not predicted by the model.

Moreover, please keep in mind that Fig. 4 shows the mean column precision for each observation whereas Fig. 2 shows a single histogram for the column precision across all measured ground cells in 80 observations. This can result in small differences when estimating percentiles from Fig. 4.

*-In the text it is said that for albedo above 0.3 the error remains below 2%, but how many of the observations are above that level of surface albedo? This links to the comment above about Figure 2 and the surface reflectance shown there.*

There are 17 observations with a mean albedo between 0.2 and 0.3 and 26 observations with an albedo greater than 0.3, which accounts for just over half of the 80 observations. We have clarified the text to emphasize that the column density precision decreases as the albedo increases.

*-How do atmospheric conditions affect the estimated precision? How does precision vary with the scattering glint angle?*

While the strongest drivers of precision are the signal levels, atmospheric conditions do affect the precision to a lesser extent via the methane Jacobian. As illustrated in the inset in Fig. 1, we sample multiple observations in space and time in order to attempt to sample a wide variety of atmospheric conditions. This clarification has been added.

As illustrated in Fig. 4a, the lower the scattering angle the higher the signal, which also corresponds to a higher surface reflectance. From Fig. 4b, the higher surface reflectance the better (lower) the column precision. Thus, the precision is better for small scattering angles, which is why we try to minimize this value when selecting the observation trigger time.

*-Paragraph line 162. There are a couple of assumptions made for which no evidence or reasoning is discussed, so I would extend that. How certain are these assumptions and how do they affect the result? Then after deriving Eq. 6, in line 183 referring to Fig. 4d you state that "you find" that the precision is linearly proportional to the scaling factor, but that has been the way it has been defined in Eq. 6. Does this then back up the assumption made, or would you expect another type of dependency?*

The paragraph in line 162 has been rephrased to emphasize this is a derivation. The assumptions in paragraph 162 are made to derive a relationship between the measured signal I, the airmass factor $\mu$, and the column precision $\Delta X_{CH_4}$. The derived relationship is shown mathematically to be linear in Eq. 6. In Fig. 4d), we verify this relationship to be indeed linear, supporting the mathematical derivation.

*Detection limit*

*-Related to the parameter q in Eq. 7, two values from Jacob et al. (2016) are mentioned without any background on what is the meaning of those. The choice of q and wind speed needs to be further discussed and justified, as the choice of q=2 and 3m/s directly impacts the detection limit.*

We reference the detection limit to 3 m/s as standard practice.

The definition of "q" in the text is as follows: "q determines the number of standard deviations above the noise required to reliably assign a methane enhancement to a pixel".

We assume, as does Jacob (2016), that a measured enhancement 2x above the noise is sufficient to detect a plume and that an enhancement 5x above the noise is sufficient for source rate quantification.

We have added a sentence to clarify this assumption.

*-There is a great discussion on how the detection limit is affected by the parameters related to geometry and latitude. However, other variables like wind speed and surface albedo are not discussed in the text.*

A direct mathematical connection to both wind speed and surface reflectance (albedo) can be made to the detection limit. The detection limit is linearly proportional to the wind speed as illustrated in Eq. 7. Should the wind speed double, the detection limit will also double. We have added a clarification after Eq. 7.

Likewise, from Eq. 6 and Eq. 9, the detection limit is found to be inversely proportional to the signal and, correspondingly, will be inversely proportional to the surface reflectivity. We have added a clarification after Eq. 9.

**Other comments:**

*Line 86: 'P (Z, Z', ws) from the satellite' reads as something the satellite measures or something that can be derived from the instrument.*

The wording has been updated.

*Line 117: in each retrieval: in each figure; the extracted plume enhancement*

Updated.

*Line 119: are these the same wind fields used in the retrieval algorithm?*

The wind fields shown are those obtained from GEOS-FP. A clarification has been added to the Figure 2 caption.

*Line 122: how representative is the wind error derived from US airports for offshore conditions? This needs to be further discussed. Offshore conditions and close to the shore wind conditions may differ from those at airports.*

As the referee points out, the wind error is derived by comparing GEOS-FP to US airports. It may be possible that GEOS-FP is more accurate over oceans due to the lack of topographical features. To answer such questions and determine how representative GEOS-FP is for offshore environments, one could compare the GEOS-FP wind products to wind data from ocean buoys. This would provide valuable insight and we believe it is worth pursuing in future work. Until such a comparison is made and analyzed, the distribution obtained from US airports provides the next best reference to the wind error.

*Please be mindful with the language: source rates are not measured but quantified (e.g., line 127), methane emission is not observed by the satellite nor shown in Fig.2f (e.g., line 129), plumes are not measured but detected.*

> These have been modified.

*Line 149: empirically measure: empirically estimate, as you estimate the precision based on empirical measurements, but you don't directly measure the precision.*

> Updated.

*Line 162 paragraph: you assume all readers are familiar with the shot-noise, but this probably not the case. I'd add a sentence explaining what that is. "In this case": unclear to which case you refer to.*

> The paragraph with Line 162 has been rewritten to clarify the assumptions made and include a short description of shot-noise.

We thank the referee for their comments and suggestions which we believe have strengthened the manuscript.